# Polypropylene Modified with Polyethylene Through Reactive Melt Blending: Fabrication and Characterizations

**DOI:** 10.3390/polym17010049

**Published:** 2024-12-28

**Authors:** Changgyu Lim, Yujin Jang, Young-Wook Chang

**Affiliations:** 1Department of Chemical Engineering, Hanyang University, Ansan 15588, Republic of Korea; freesuu@hanyang.ac.kr (C.L.); serapina1027@hanyang.ac.kr (Y.J.); 2BK21 FOUR ERICA-ACE Center, Hanyang University, Ansan 15588, Republic of Korea

**Keywords:** PP modified with PE, reactive melt blending, mechanical properties, rheological properties, microcellular foam

## Abstract

Conventional PP with a linear chain structure is not suitable for foam processing due to its poor rheological properties. In this study, PP was modified with PE through reactive melt blending of maleic anhydride-grafted PP (MA-PP) with a small amount of PE bearing glycidyl groups on its backbone (G-PE), with the aim of enhancing the melt rheological properties of PP to make it suitable for foam processing. An anhydride–epoxy reaction occurred between MA-PP and G-PE during the melt processing, resulting in the formation of a crosslinked polymer network, which was confirmed by FTIR spectroscopy, a solubility test, and the presence of a rubbery plateau above the melting point. Melt rheological tests demonstrated that the modified PP showed a pronounced shear-thinning effect and higher elasticity compared to pristine PP. Foaming tests using supercritical carbon dioxide as a foaming agent in an autoclave demonstrated that the modified PP could produce a microcellular foam with a closed-cell structure, which was not achievable with neat PP.

## 1. Introduction

Polypropylene (PP), along with polyethylene (PE), is a widely used thermoplastic in modern society due to its low density, excellent mechanical properties, high resistance to water and chemicals, and cost-effectiveness [1,2,3,4,5,6]. However, the rheological properties of conventional PP resins are not suitable for advanced melt processing applications such as foaming, extrusion coating, and blow molding, primarily due to their inherently linear chain structure [7,8,9].

To overcome this limitation, various reactive melt processing methods have been explored to modify the molecular architecture of PP, including the introduction of long chain branching or crosslinking. Conventional methods to endow improved rheological properties to PP typically involve radical-induced reactions through peroxide decomposition [10,11,12,13,14,15] or high-energy irradiation [16,17,18,19]. However, these approaches often lead to chain scission and uncontrolled molecular modifications, leading to the deterioration of the mechanical properties of PP.

Multi-step reactive melt processing has been regarded as an effective way to improve the rheological properties of PP while preserving its mechanical properties [20,21,22,23,24,25,26,27,28,29]. This method typically involves grafting maleic anhydride onto PP, followed by further modification of the maleic anhydride-functionalized PP (MA-PP) using various reactive agents [20,21,22,23,24,25,26,27,28,29]. For instance, Tang et al. [22] and Guapacha et al. [23] demonstrated that the MA-PP could be crosslinked using low-molecular-weight epoxy as a crosslinker, while Létoffé et al. utilized multifunctional amine compounds for crosslinking MA-PP [24,25]. Other studies, such as those by Li et al. [26] and Maroofy et al. [27], showed that MA-PP can be modified to have ionically crosslinked structure through zinc neutralization through the addition of metals. More recently, Muljana et al. converted the maleic anhydride in MA-PP into other reactive groups, such as furan or imidazole, followed by crosslinking with bismaleimide [28,29]. Although these methods improved the melt strength and elasticity of PP, some challenges remain including poor compatibility of the crosslinkers with MA-PP and environmental concerns due to the release of volatile organic compounds (VOCs) during the melt processing.

Polymer blending has been explored as a complementary strategy for achieving enhanced rheological properties in an economical manner [30,31,32,33,34]. Yamaguchi et al. reported that PP/LDPE blends show an improved melt strength over neat PP [31]. Additionally, Lopez-Barron et al. showed that incorporating a PP-PE comb block copolymer as a compatibilizer further enhanced the melt strength of PP/PE blends [32]. Although use of PE as a modifier provides a scalable, cost-effective, and practical way for producing high-performance PP suitable for advanced applications such as lightweight foamed materials, studies on this approach are rare.

In this study, PP was modified with PE through reactive melt blending of maleic anhydride-grafted PP (MA-PP) with a small amount of PE bearing glycidyl groups on its backbone (G-PE), with the aim of improving the melt rheological properties of PP to enhance its suitability for foam processing while maintaining its superior mechanical properties. During melt mixing, the glycidyl groups on G-PE react with the maleic anhydride groups on MA-PP, leading to a crosslinked structure. This modification enables the formation of microcellular foams with a closed-cell structure, which cannot be achieved with neat PP alone. The findings and details of this process are presented in this paper.

## 2. Materials and Methods

### 2.1. Materials and Sample Preparation

Maleated polypropylene (MA-PP; maleic anhydride content: 1 wt%; Adpoly BP402) was procured from Lotte Chem (Daejeon, Korea). Poly(ethylene-co-glycidyl methanacrylate) (G-PE; glycidyl content: 9 wt%), zinc acetylacetonate (Zn(acac)_2_) and xylene were purchased from Sigma-Aldrich (St. Louis, MO, USA).

MA-PP and G-PE were melt-mixed so that the G-PE content was up to 20 wt % in the presence of 1.0 phr of Zn(acac)_2_ using a Haake mixer (Haake Polylab Rheomix 600, Thermo Fisher Scientific, Bremen, Germany) for 20 min at 180 °C with a rotor speed of 60 rpm. The obtained mixtures were compression-molded into films with a uniform thickness of about 1 mm using a hot press at 180 °C for 10 min.

### 2.2. Characterization

An FTIR analysis was carried out with 32 scans and a resolution of 4 cm^−1^ using a Bucker ALPHA spectrometer (Nicolet IS10, Thermo scientific, Waltham, MA, USA) equipped with an attenuated total reflectance (ATR) accessory, under a nitrogen atmosphere to examine the reaction between maleic anhydride in the MA-PP and epoxy in the G-PE.

To determine whether the sample formed a chemically crosslinked structure, a solubility test was performed using xylene as the solvent at 120 °C.

Dynamic mechanical tests were carried out by using a dynamic mechanical analyzer (TA Instrument, model DMA-Q800, New Castle, DE, USA). The samples were subjected to cyclic tensile strain with an amplitude of 0.2% at a frequency of 1 Hz. The temperature was increased at a heating rate of 10 °C/min over the range of −100 to 200 °C.

Melting transitions and the accompanying enthalpy changes of the samples were examined using a differential scanning calorimeter (TA instruments, DSC Q20 equipped with RSC90, a refrigerated cooling system; New Castle, DE, USA) under a nitrogen gas flow. A 10 mg sample was heated from room temperature to 200 °C at a rate of 10 °C/min and was kept for 5 min at this temperature. Then, the sample was cooled down to −50 °C at a cooling rate of 10 °C/min followed by reheating it to 200 °C at a rate of 10 °C/min to monitor the crystallization and crystalline melting transition of the sample, respectively.

The tensile test was conducted using a universal testing machine equipped with a 1 kN load cell (UTM, AGS-500NX, Shimadzu, Tokyo, Japan) with a crosshead speed of 50 mm/min at room temperature. The tests were repeated until reproducible results were obtained.

The melt rheological properties were examined using a torsional parallel plate rheometer (MCR 300, Anton Paar, Graz, Austria) with a 2 mm gap between the plates, under the small-amplitude oscillatory shear mode at 180 °C. A frequency sweep was carried out within a frequency range of 0.1–100 rad/s at a strain amplitude of 1% within the linear viscoelastic range.

Foam processing was performed using supercritical carbon dioxide (CO_2_) as a blowing agent. A polymer sample was saturated with CO_2_ at 150 °C under pressure of 30 MPa in an autoclave connected to a CO_2_ cylinder, and then the pressure was reduced rapidly to atmospheric conditions so that the CO_2_ gas in the polymer could expand. The system was maintained at atmospheric pressure for approximately 1 h so that the bubbles could complete their growth. The cell morphology of the foamed samples was examined using scanning electron microscopy ((FE-SEM, S-900, Hitachi Co., Tokyo, Japan) and the average cell size *d* was determined from the SEM images. The cell density (*N*_c_) (the number of cells per unit volume of foam) was determined from following equation [35,36,37]:*N*_c_ (cells/cm^3^) = (n/A)^3/2^ × (ρ/ρ_f_) 
where n is the number of cells in the micrograph of area A (in cm^2^), and ρ and ρ_f_ are the density of the sample before and after the foaming, respectively.

## 3. Results and Discussion

### 3.1. FTIR Analysis and Solubility Test

To verify that the anhydride–epoxy reaction occurred in the MA-PP/G-PE blend, an FTIR analysis was conducted, and the results are shown in Figure 1a. For the blend samples, the MA-PP/G-PE (80/20) blend was selected for the FTIR analysis because, according to preliminary observations, it exhibited distinctive features that were comparable to those of the sample with a lower G-PE content. The FTIR spectra of neat MA-PP exhibited characteristic absorption bands at 1787 cm^−1^ and 1863 cm^−1^, corresponding to the stretching vibrations of maleic anhydride [38,39]. In the FTIR spectra of neat G-PE, the absorption bands at 846 cm^–1^ and 910 cm^−1^ were attributed to the stretching vibrations of the epoxy ring [40,41]. In the FTIR spectrum of the MA-PP/G-PE (80/20) blend, these peaks were absent, and a new absorption band appeared at 1733 cm^−1^ which can be ascribed to the C=O stretching of ester bonds formed by the reaction between the epoxy group of G-PE and the maleic anhydride groups of MA-PP [22,23]. Figure 1b illustrates the anhydride–epoxy reaction between MA-PP and G-PE.

The solubility test provided complementary evidence of the chemical crosslinking in the MA-PP/G-PE blend; the results are shown in Figure 2. Both neat MA-PP and G-PE dissolved completely in xylene at 120 °C due to their linear structure, whereas the as-prepared MA-PP/G-PE blend did not dissolve but swelled under the same conditions. This result confirmed that the MA-PP/G-PE blend formed a chemically crosslinked network through the anhydride–epoxy reaction between the two polymers.

### 3.2. Dynamic Mechanical Properties

Figure 3a,b shows the variation in the storage modulus (*E’*) and tan δ of the samples with temperature, respectively. In Figure 3a, there is a sudden drop in the modulus at the melting point in neat MA-PP and neat G-PE, which is a typical behavior for linear thermoplastics. In contrast, the blend displayed a rubbery plateau above the melting point. This rubbery plateau, characteristic of crosslinked polymers, confirmed the formation of a crosslinked network in the MA-PP/G-PE blend [23,24]. The *E’* value of the blend samples at the plateau region (at 150 °C; Table 1), which is correlated with the degree of crosslinking, increased from 0.026 to 0.054 when the G-PE content increased from 5 to 20 wt%.

The glass transition temperature (*T*_g_) of the samples obtained from Figure 3b are shown in Table 1. The *T*_g_ values of neat MA-PP and neat G-PE were approximately 0 °C and −47.2 °C, respectively. In the crosslinked MA-PP/G-PE blends, the *T*_g_ of the PP phase shifted to a higher temperature, increasing by around 0.7 °C and 1.5 °C when the G-PE content was 10 wt% and 20 wt%, respectively, while the *T*_g_ of the PE phase in the blend was not clearly detected. The observed increase in the *T*_g_ of the modified PP compared to neat PP indicates that the mobility of the PP chain is restricted due to the formation of a crosslinked structure [23,26,27].

### 3.3. Melting and Crystallization Behavior

The melting and crystallization behavior of the samples was analyzed using DSC, and the thermograms are shown in Figure 4b. The melting temperature (*T*_m_) and crystallization temperature (*T*_c_) values, as determined from the thermograms, are summarized in Table 2. The melting endotherm of neat MA-PP was observed over a wide temperature range, from approximately 100 °C to 140 °C, while that of neat G-PE appeared around 104 °C. Consequently, the melting endotherm of the PP phase overlapped with that of the G-PE phase in the blend, making it difficult to precisely determine the melting enthalpy of each component polymer in the blends. The melting endotherm peak of the G-PE phase in the blends could only be clearly identified when the G-PE content reached 20 wt%. As a result, only the *T*_m_ values of the samples are reported in Table 2. The results show that the *T*_m_ of the PP and PE phases in the MA-PP/G-PE (80/20) blend decreased by approximately 1.6 °C and 3.7 °C, respectively, compared to neat MA-PP and neat G-PE. This reduction suggests that the crystallization of both component in the blends was hindered due to the reduced molecular mobility of the polymer chains in the crosslinked network.

The melt crystallization thermograms shown in Figure 4b indicate that the crystallization temperature of MA-PP shifted slightly to higher values (by approximately 1.0–1.6 °C) in the blends compared to that of neat MA-PP when the G-PE content was less than 10 wt%. This increase in *T*_c_ for the blend is likely due to the nucleating effect of long-chain branches formed on the PP chain during the crosslinking of MA-PP by G-PE, which has also been observed in long-chain branched PP [22,23]. However, in the blend containing 20 wt% G-PE, the increase in *T*_c_ was marginal, suggesting that the high degree of crosslinking in the blend, which was confirmed by the dynamic mechanical analysis, suppresses the nucleating effect.

### 3.4. Tensile Properties

Figure 5 presents the stress–strain curves for all the samples, and the Young’s modulus, yield stress, tensile strength, and elongation-at-break determined from the curves are summarized in Table 3. The neat MA-PP showed ductile deformation with a large elongation, yielding, and cold drawing at high strain levels. The modified PP exhibited similar ductile deformation, but with a lower Young’s modulus and a reduced elongation-at-break compared to neat MA-PP. The decrease in Young’s modulus with increasing PE content was attributed to the lower rigidity of PE relative to PP. The reduction in elongation-at-break was related to the crosslink density, as was observed in the dynamic mechanical analysis discussed earlier. Interestingly, the blends showed a higher yield stress and tensile modulus beyond the yield point compared to the neat component polymers. These synergistic effects in the tensile properties are likely associated with strong interfacial bonding between the component polymers formed by the epoxy–anhydride reaction and the phase-separated morphology of the blends. For a comprehensive understanding of this behavior, however, further studies are necessary and will be explored in future research.

### 3.5. Rheological Properties

The melt rheological behavior of the PP modified with PE was examined using an oscillating rheometer, and the results are displayed in Figure 6. In Figure 6a, it is obvious that the MA-PP/G-PE blends showed a higher complex viscosity in the low frequency region compared to the neat component polymers, and it increased with increasing G-PE content. This is because the blends formed a crosslinked network and the degree of crosslinking increased with increasing G-PE content, as confirmed in the previous section. It should also be noted that the blends showed more pronounced shear-thinning effects compared to the neat polymers. This increase in the shear-thinning effect has been observed in long-chain branched or crosslinked polymers [23,26,28,42,43,44,45].

Figure 6b shows the storage modulus (G’) and loss modulus (G’’) of the neat MA-PP and MA-PP/G-PE blend samples as a function of frequency. It can be observed that the G’ and G’’ of the blends were higher than those of neat MA-PP, especially in the low frequency region, accompanied by a decrease in the terminal region slope. This lower terminal slope and higher modulus indicates a solid-like behavior of the modified PP, which originated from the crosslinked structure of the blends. It should also be noted that the G’’ was larger than the G’ in entire frequency region in neat MA-PP, which had a liquid-like behavior, whereas the opposite was observed in the MA-PP/G-PE (80/20) blend in the entire frequency zone indicating that it had a solid-like behavior [25,26,27,28,29]. This trend is in accordance with the degree of crosslinking of the samples.

The loss tangent (tan δ = G’’/G’) of neat MA-PP and the MA-PP/G-PE blends are displayed in Figure 6c. As can be seen in the figure, the blends had a much lower tan δ compared to neat MA-PP in the low frequency region, indicating that the blend samples have a higher melt elasticity [28,29].

### 3.6. Foam Processing

Figure 7 shows SEM micrographs of the freeze–fracture surfaces of the foams obtained from neat MA-PP and the MA-PP/G-PE blends. In the case of the pristine MA-PP foam and MA-PP/G-PE (95/5) blend foam, the cells were not uniform and some cells had open structures, implying that the melt strength of these polymers was not sufficiently high enough to prevent coalescence of the cells during the foaming.

The cell density, N_c_, of these foams was 7.0×10^6^ and 8.5×10^7^ cells/cm^3^, respectively. In contrast, the foams produced from the MA-PP/G-PE (90/10) and (80/20) blends exhibited a closed-cell structure with improved uniformity in cell size. The average cell size of these foams were 17.8 µm and 15.2 µm, respectively. The cell density of these foams increased to 6.72 × 10^8^ and 1.04 × 10^9^ cells/cm^3^, respectively, which was about two orders of magnitude higher than that of the pure MA-PP foam. The improved uniformity in cell size and marked increase in the cell density indicate that these blend samples had a high melt strength that could withstand the elongational strain imposed on the polymer during the expansion of the gas in the molten polymer [26,33,34].

The microcellular foam with a closed-cell structure fabricated using the modified PP can have a wide range of applications, such as packaging materials, automotive components, and construction materials for thermal and sound insulation due to its cost-effectiveness, light weight characteristics, good resistance to moisture, as well as good recyclability and higher service temperature.

## 4. Conclusions

This study demonstrated that polypropylene (PP) with enhanced rheological properties that are suitable for foam production can be effectively fabricated through reactive melt blending of two different commercially available polyolefins with reactive groups: maleic anhydride-functionalized PP (MA-PP) with a small amount of polyethylene (PE) containing glycidyl moieties on its backbone (G-PE). The formation of a crosslinked structure in the modified PP, driven by the anhydride–epoxy reaction, was confirmed, resulting in significant improvements in its melt rheological properties. Foaming tests using supercritical CO_2_ as a blowing agent further revealed that the modified PP is highly suitable for producing microcellular foam with a closed-cell structure.

While this research aimed to address the current challenges associated with conventional polymers, we also emphasize that ongoing efforts are directed toward exploring biopolymers and other sustainable materials for foam production, aligning with the global trends in sustainability.

## Figures and Tables

**Figure 1 polymers-17-00049-f001:**
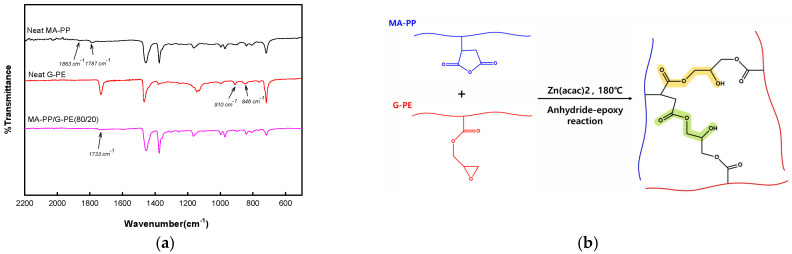
(**a**) FT-IR spectra. (**b**) Anhydride–epoxy reaction in MA-PP/G-PE blend.

**Figure 2 polymers-17-00049-f002:**
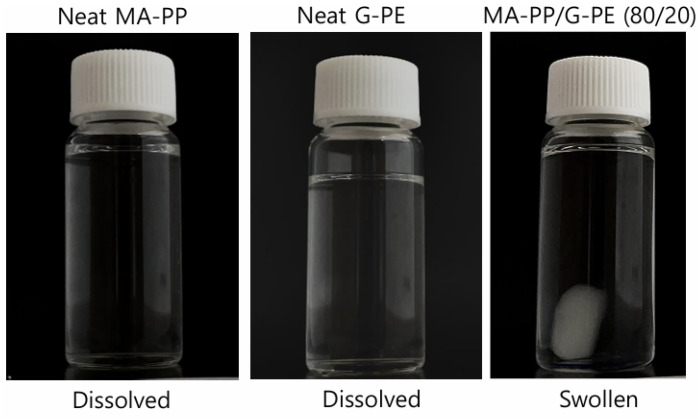
Solubility test.

**Figure 3 polymers-17-00049-f003:**
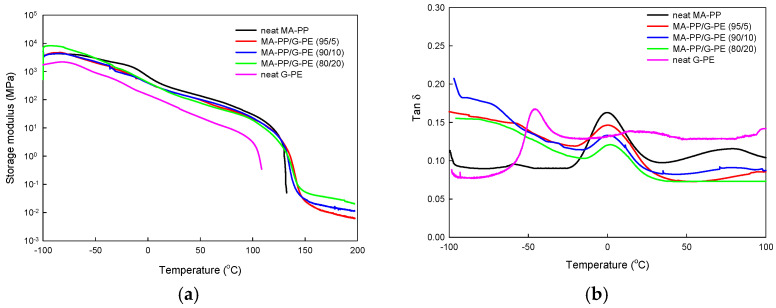
Variation in (**a**) dynamic storage modulus and (**b**) tan δ of samples with temperature.

**Figure 4 polymers-17-00049-f004:**
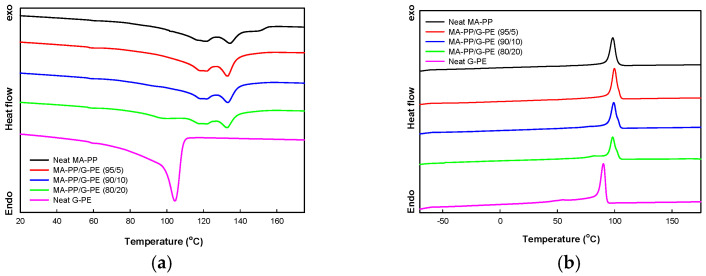
DSC thermograms of samples. (**a**) Heating scan; (**b**) cooling scan.

**Figure 5 polymers-17-00049-f005:**
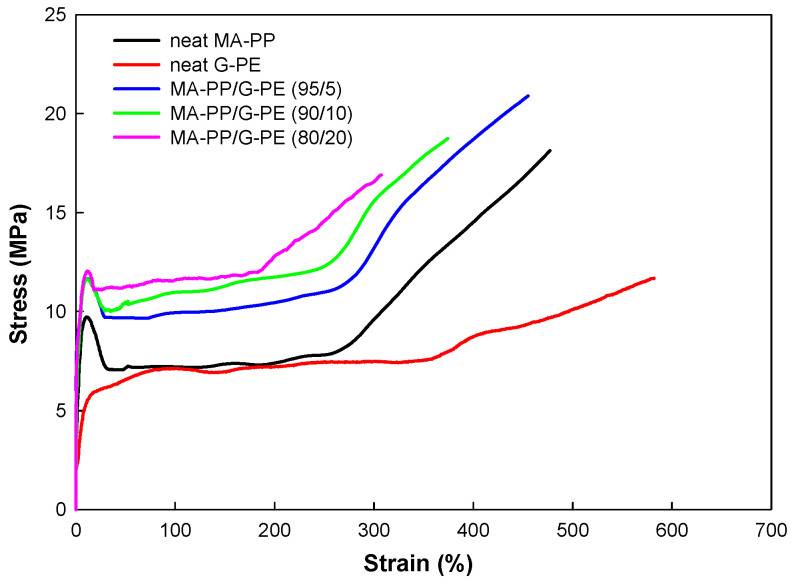
Stress–strain curves of samples.

**Figure 6 polymers-17-00049-f006:**
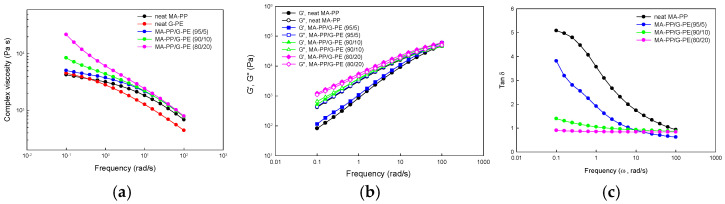
Variation in (**a**) complex viscosity, (**b**) storage/loss modulus, and (**c**) tan δ with angular frequency of samples.

**Figure 7 polymers-17-00049-f007:**
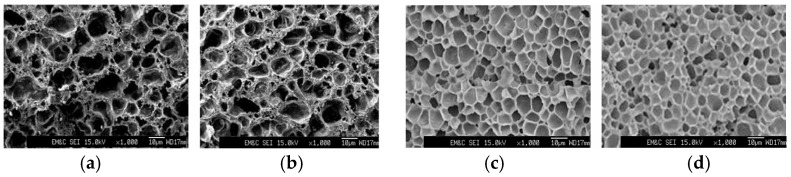
Cell morphology of foams produced from (**a**) neat MA-PP, (**b**) MA-PP/G-PE (95/5), (**c**) MA-PP/G-PE (90/10), and (**d**) MA-PP/G-PE (80/20) blends.

**Table 1 polymers-17-00049-t001:** Storage modulus (*E’*) at rubbery plateau region (*T* = 150 °C) and *T*_g_ obtained from tan δ peak maximum of samples.

Sample	*E’* at 150 °C (MPa)	*T*_g_ of MA-PP (°C)	*T*_g_ of G-PE (°C)
Neat MA-PP	0	−0.04	
Neat G-PE	0	-	−47.2
MA-PP/G-PE			
95/5	0.026	0.21	-
90/10	0.031	0.65	-
80/20	0.054	1.41	-

**Table 2 polymers-17-00049-t002:** Thermal characteristics of samples.

Sample	*T*_m_ of MA-PP (°C)	*T*_m_ of G-PE (°C)	*T*_c_ of MA-PP (°C)
Neat MA-PP	121.8, 134.4	-	98.1
Neat G-PE	-	104.4	-
MA-PP/G-PE			
95/5	121.8, 133.2	-	99.7
90/10	121.8, 133.0	-	99.2
80/20	121.9, 132.8	100.7	98.3

**Table 3 polymers-17-00049-t003:** Tensile properties of samples.

Sample	Young’s Modulus (MPa)	Yield Stress (MPa)	Tensile Strength (MPa)	Elongation-at-Break (%)
Neat MA-PP	220 ± 4.2	9.5 ± 0.1	18.1 ± 0.2	480 ± 18
Neat G-PE	77 ± 2.1	-	11.7 ± 0.1	580 ± 22
MA-PP/G-PE				
95/5	215 ± 3.3	11.6 ± 0.1	20.9 ± 0.2	450 ± 18
90/10	210 ± 3.6	11.6 ± 0.1	18.7 ± 0.2	370 ± 12
80/20	215 ± 3.5	12.0 ± 0.1	16.9 ± 0.1	310 ± 10

## Data Availability

Data are contained within the article.

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
