# Peer review of "Polypropylene Modified with Polyethylene Through Reactive Melt Blending: Fabrication and Characterizations"

_polymers, 2024, doi:10.3390/polym17010049_

Round 1
Reviewer 1 Report
Comments and Suggestions for Authors
The manuscript presented an approach to blend maleic anhydride-functionalized polypropylene (MA-PP) with epoxy-containing polyethylene (G-PE), forming a cross-linked polymer network, possibly due to the chemical reaction between the anhydride and epoxy groups. Before publication, the manuscript needs revisions:
>The abstract does not make it clear why PP/PE foams are being produced. Why not just PP or PE? The authors should quantify the gain in properties and indicate the best composition.
>Introduction. The manuscript highlights the use of commodity polymers (PP and PE) for foam production. Why did it not address new environmentally friendly materials (biopolymers)? This is relevant considering sustainable trends.
>Page 2 and 3. Some procedure descriptions should be added with more details. For example, for FTIR, what is the scan and resolution? The solubility test should follow the methodology presented in characterization; In DSC, what was the gas flow and mass used?; What is the load cell value for the tensile test? What is the gap between the plates in the rheology test? Please report in the manuscript;
>Results and discussion. The authors processed the materials in an internal mixer, which results in a torque vs time curve. Why was the result not added to evaluate reactivity?
>Page 3. “characteristic absorption bands were observed at 1787cm–1 and 1863cm–1…..”. Please add literature reports indicating the chemical groups of mentioned bands; Improve the written part in the reaction of Figure 1, the words are blurred;
> What was the motivation for selecting the 80/20% composition for FTIR? It is not clear from the manuscript. Please indicate at the beginning of the results and discussion whether it was based on previous reports;
>Page 4. “This result indicates that the MA-PP/G-PE blend forms a chemically crosslinked network through a anhydride-epoxy reaction between the two polymers.” Correlate the solubility result with the FTIR presented previously;
> Variation of storage modulus with temperature. Do the authors not have the Tan (delta) curve? This could evaluate the behavior of the glass transition of PP between the range of -30-0°C;
Author Response
Comments #1 : The abstract does not make it clear why PP/PE foams are being produced. Why not just PP or PE? The authors should quantify the gain in properties and indicate the best composition.
[Answers] Thank you for the comment. The primary goal of this work is to enhance the rheological properties of PP to make it more suitable for foam processing applications. Neat PP has limitations in foamability due to its poor rheological properties, while these properties are significantly enhanced when PP was modified with a small amount of PE. This modification enables the formation of microcellular foams with a closed-cell structure, which cannot be achieved with neat PP alone. The abstract has been revised to clarify these points.
Comments #2 : Introduction. The manuscript highlights the use of commodity polymers (PP and PE) for foam production. Why did it not address new environmentally friendly materials (biopolymers)? This is relevant considering sustainable trends.
[Answers] Thank you for your insightful question. Our manuscript focuses on PP due to their widespread industrial application, cost-effectiveness, and well-established processing techniques. The primary objective of this study was to improve melt rheological properties of PP to enhance its suitability for the foam production processes. Addressing these properties is crucial for optimizing foam quality and expanding the applications of these commodity polymers. Future work will certainly consider exploring the use of biopolymers and other sustainable materials to align with global trends in sustainability.
Following sentences were added to Conclusion. “This research aimed to address current challenges associated with conventional polymers. However, we would like to emphasize that we are actively exploring the use of biopolymers and other sustainable materials for foam production to align with global sustainability trends.”
Comments #3 : Page 2 and 3. Some procedure descriptions should be added with more details. For example, for FTIR, what is the scan and resolution? The solubility test should follow the methodology presented in characterization; In DSC, what was the gas flow and mass used?; What is the load cell value for the tensile test? What is the gap between the plates in the rheology test? Please report in the manuscript.
[Answers] Thanks for the valuable suggestion. The experimental procedure and test conditions were added as per the comments.
Comments #4 : Results and discussion. The authors processed the materials in an internal mixer, which results in a torque vs time curve. Why was the result not added to evaluate reactivity?
[Answers] Thanks for the valuable suggestion. We would like to inform you that monitoring the variation of torque during the mixing process was not possible using the internal mixer in our laboratory. This was due to a malfunction in the mixer's torque sensor, which is currently not operational.
Comments #5 : Page 3. “characteristic absorption bands were observed at 1787cm–1 and 1863cm–1…..”. Please add literature reports indicating the chemical groups of mentioned bands; Improve the written part in the reaction of Figure 1, the words are blurred;
[Answers] Thanks for the valuable comment. The explanation for Figure 1 has been carefully revised, and the related references have been included.
Comment #6 : What was the motivation for selecting the 80/20% composition for FTIR? It is not clear from the manuscript. Please indicate at the beginning of the results and discussion whether it was based on previous reports;
[Answers] Thanks for the comment. The MA-PP/G-PE (80/20) composition was selected for FTIR analysis because, according to preliminary observations, it exhibited distinctive features clearly compared to the sample with lower G-PE content. This was remarked in the revised manuscript.
Comment #7 : Page 4. “This result indicates that the MA-PP/G-PE blend forms a chemically crosslinked network through a anhydride-epoxy reaction between the two polymers.” Correlate the solubility result with the FTIR presented previously;
[Answers] Thank you for the comment. The solubility test and FTIR analysis provide complementary evidence of the chemical crosslinking in the MA-PP/G-PE blend. The FTIR analysis confirmed the occurrence of the anhydride-epoxy reaction, which led to the formation of ester bonds between the two polymers, resulting in chemical crosslinking in the blend. The solubility test further supports this conclusion, as the MA-PP/G-PE blend, unlike the neat polymers, did not dissolve in xylene but instead swelled, indicating the formation of a chemically crosslinked network. Together, these results demonstrate that the anhydride-epoxy reaction between the two polymers leads to the creation of a crosslinked structure.
Comment #8 : Variation of storage modulus with temperature. Do the authors not have the Tan (delta) curve? This could evaluate the behavior of the glass transition of PP between the range of -30-0°C;
[Answers] Tan delta curve was added as Figure 3(b) and glass transition temperature obtained from the curve was summarized in Table 1, The discussions for the result was added.
Reviewer 2 Report
Comments and Suggestions for Authors
Polypropylene was modified with polyethylene via reactive melt blending of maleic an- hydride-graft-polypropylene. The structures, mechanical, and rheological properties of the modified PPs were investigated, and their foamability was also examined using supercritical carbon dioxide as a blowing agent.
This manuscript can be accepted after moderate revision.
1. The format of references is awful.
2. Please check for any grammatical or spelling errors in the whole manuscript.
3. What is the novelty of this work? The authors should highlight the innovation of this research.
4. What is the main application field of polyethylene modified polypropylene? Or the authors should give more explanation on the performance and properties of the fabricated samples.
5. In the experimental part, why did the authors use Zn(acac)2?
6. The authors modified the polypropylene with up to 20wt% polyethylene. Why 20wt% polyethylene? Why not 10wt% polyethylene or 30wt% polyethylene?
7. For the tensile properties, the authors should give more explanation. In addition, the authors should demonstrate the reasons for the modified samples presenting better properties than neat MA-PP and G-PE.
Author Response
Comments #1 : The format of references is awful.
[Answers] Thanks for the comment. Format of references was corrected.
Comments #2: Please check for any grammatical or spelling errors in the whole manuscript.
[Answers] Thanks for the comment. Grammatical and spelling errors were corrected in the whole manuscript.
Comments #3: What is the novelty of this work? The authors should highlight the innovation of this research.
[Answers] Thanks for the comment. The novelty of this study lies in demonstrating that the rheological properties of maleic anhydride-modified polypropylene (MA-PP) can be successfully enhanced through reactive melt blending with a small amount of polyethylene bearing glycidyl moieties (G-PE), a commercially available functionalized polyethylene. This modification enables the production of microcellular foam with a closed-cell structure, which cannot be achieved with neat PP. G-PE provides advantages over low molecular weight reactive modifier reported in the literature, such as epoxy, amines, or metals, which are often associated with issues like poor compatibility with MA-PP and environmental concerns due to volatile organic compounds (VOCs). Moreover, the incorporation of a minimal amount of G-PE preserves the superior intrinsic properties of polypropylene, making this approach both effective and practical. The novelties and innovation of this work was emphasized in the revised manuscript.
Comments #4 : What is the main application field of polyethylene modified polypropylene? Or the authors should give more explanation on the performance and properties of the fabricated samples.
[Answers] As mentioned in the reply to comment #3, the modified PP prepared in this study retains the superior properties of PP due to the use of only a small amount of PE for the modification. Foam manufactured using this modified PP possesses several advantageous properties, including lightweight characteristics, high service temperature, good resistance to chemicals and moisture. These characteristics make the foam suitable for a wide range of applications, such as lightweight and durable packaging materials for protecting goods, automotive components like interior parts and insulation, and construction materials for thermal and sound insulation. Furthermore, it is recyclable as a thermoplastic, enhancing its environmental friendliness.
Comments #5. In the experimental part, why did the authors use Zn(acac)2?
[Answer] Thanks for the comment. Zn(acac)â‚‚ (zinc acetylacetonate) is a well-known catalyst that enhances the anhydride-epoxy reaction rate and efficiency as well as facilitates the transesterification reaction of β-hydroxy ester bonds within the crosslinked network formed by the anhydride-epoxy reaction.
Comments #6 : The authors modified the polypropylene with up to 20wt% polyethylene. Why 20wt% polyethylene? Why not 10wt% polyethylene or 30wt% polyethylene?
[Answers] Thank you for your comment. PE has lower modulus and service temperature compared to PP. Therefore, we aimed to modify the PP with the minimal amount of PE necessary to achieve the desired rheological properties for fabricating microcelluar foam with closed cell structure. Desired foam structure was achieved when the PP was modified with up to 20 wt% PE, leading us to conclude that incorporating more than 20 wt% PE is unnecessary.
Comments #7: For the tensile properties, the authors should give more explanation. In addition, the authors should demonstrate the reasons for the modified samples presenting better properties than neat MA-PP and G-PE.
[Answers] Thanks for the comment. Explanations for the tensile properties were added. At present, however, we could not provide clear explanations why the modified PP present higher yield stress than neat MA-PP and G-PE sufficiently. It is likely to be associated with strong interfacial bonds between the component polymers in the blends and their phase-separated morphology. This will be explored in the further study.
Round 2
Reviewer 1 Report
Comments and Suggestions for Authors
The authors have commented on all questions satisfactorily. At the same time, recommendations have been added to the new version of the manuscript, improving its quality. The manuscript has merit for publication in Polymers.